# Evaluation of the Use of Singleplex and Duplex CerTest VIASURE Real-Time PCR Assays to Detect Common Intestinal Protist Parasites

**DOI:** 10.3390/diagnostics14030319

**Published:** 2024-02-01

**Authors:** Alejandro Dashti, Henar Alonso, Cristina Escolar-Miñana, Pamela C. Köster, Begoña Bailo, David Carmena, David González-Barrio

**Affiliations:** 1Parasitology Reference and Research Laboratory, National Centre for Microbiology, Health Institute Carlos III, 28220 Majadahonda, Spain; dashti.alejandro@gmail.com (A.D.); pamelakster@yahoo.com (P.C.K.); begobb@isciii.es (B.B.); david.gonzalezb@isciii.es (D.G.-B.); 2Department of Microbiology, Paediatrics, Radiology, and Public Health, Faculty of Medicine, University of Zaragoza, 50009 Saragossa, Spain; 3Department of Animal Production and Food Science, Faculty of Veterinary, University of Zaragoza, 50013 Saragossa, Spain; 4Faculty of Health Sciences, Alfonso X El Sabio University (UAX), 28691 Villanueva de la Cañada, Spain; 5Faculty of Medicine, Alfonso X El Sabio University (UAX), 28691 Villanueva de la Cañada, Spain; 6CIBER Infectious Diseases (CIBERINFEC), Health Institute Carlos III, 28008 Madrid, Spain

**Keywords:** molecular diagnostics, real-time PCR, gastrointestinal parasites, diarrhoea, microbiology laboratory

## Abstract

*Cryptosporidium* spp., *Giardia duodenalis* and *Entamoeba histolytica* are species of protozoa- causing diarrhoea that are common worldwide, while *Entamoeba dispar*, *Dientamoeba fragilis* and *Blastocystis* sp. appear to be commensal parasites whose role in pathogenicity remains controversial. We conducted the clinical evaluation of five singleplex and one duplex CerTest VIASURE Real-Time PCR Assays against a large panel of positive DNA samples (*n* = 358), and specifically to *Cryptosporidium* spp. (*n* = 96), *G. duodenalis* (*n* = 115), *E. histolytica* (*n* = 25) *E. dispar* (*n* = 11), *Blastocystis* sp. (*n* = 42), *D. fragilis* (*n* = 37), and related parasitic phylum species such as Apicomplexa, Euglenozoa, Microsporidia and Nematoda. DNA samples were obtained from clinical stool specimens or cultured isolates in a national reference centre. Estimated diagnostic sensitivity and specificity values were 0.94–1 for *Cryptosporidium* spp., 0.96–0.99 for *G. duodenalis*, 0.96–1 for *E. histolytica*, 1–1 for *E. dispar*, and 1–0.99 for *D. fragilis* in the evaluated singleplex assays. In the duplex assay for the simultaneous detection of *Blastocystis* sp. and *D. fragilis* these values were 1–0.98 and 1–0.99, respectively. Measures of diagnostic precision for repeatability and reproducibility were found to be under acceptable ranges. The assays identified six *Cryptosporidium* species (*C. hominis*, *C. parvum*, *C. canis*, *C. felis*, *C. scrofarum*, and *C. ryanae*), four *G. duodenalis* assemblages (A, B, C, and F), and six *Blastocystis* subtypes (ST1-ST5, and ST8). The evaluated singleplex and duplex VIASURE Real-Time PCR assays provide sensitive, practical, and cost-effective choices to the molecular diagnosis of the main diarrhoea-causing intestinal protists in clinical microbiology and research laboratories.

## 1. Introduction

Intestinal protozoa, including *Cryptosporidium* spp., *Entamoeba histolytica*, and *Giardia duodenalis*, contribute to the global burden of diarrhoeal diseases and are notably relevant to public health [1,2]. These pathogens are responsible for a wide range of gastrointestinal manifestations that lead to acute and chronic diarrhoea, abdominal pain, malnutrition, and diverse long-term sequelae [3,4,5]. In addition, impaired linear growth and cognition have been documented in young children, particularly those living in resource-poor settings [6,7]. Both *Cryptosporidium* spp. and *G. duodenalis* are also major causes of waterborne and foodborne gastrointestinal disease outbreaks that are mostly detected in medium- to high-income countries [8,9]. Not surprisingly, *Cryptosporidium* spp., *G. duodenalis*, and *E. histolytica* are responsible for seven out of ten gastrointestinal parasites that are annually diagnosed in European clinical settings [10]. Although *Entamoeba dispar* is generally considered non-pathogenic, its common presence in human stool samples can, because of its morphological similarity to pathogenic *E. histolytica*, impair diagnostic efforts [11]. *Blastocystis* sp. and *Dientamoeba fragilis* are other intestinal protists that have been recognized as potential contributors to intestinal and extraintestinal manifestations [12,13,14], although the fact that both microorganisms are often found in individuals without obvious clinical manifestations complicates their consideration as primary pathogens.

Molecular assays based on real-time PCR (qPCR) are rapidly replacing conventional detection methods (e.g., microscopy) in the first line diagnosis of diarrhoea-causing intestinal protists that have a tendency to move rapidly from detection of a single pathogen to detection of multiple pathogens. This tendency is particularly evident in medium- to high-income countries with well-equipped clinical laboratories, where the gastrointestinal disease burden of protists is low and diagnostic sensitivity is an issue [15,16]. The benefits of qPCR include:(i) high throughput of stool screening, (ii) simultaneous detection of multiple pathogens within a single sample, and (iii) fast and accurate generation of results, enabling timely clinical decisions and swift interventions [17,18,19,20]. In resource allocation, qPCR assays minimize the need for repeated sample processing and analysis, reduce turnaround times, and improve workflows and operational efficiency in a cost-effective manner [21,22].

The validation of new commercially available diagnostic assays is one of the main tasks performed by national reference centres, which can bring together technical resources (e.g., biological samples for reference purposes, equipment) and expertise to perform the task efficiently. Here, we evaluated the clinical diagnostic performance of five singleplex and one duplex VIASURE Real-Time PCR assays in the detection and identification of common and clinically relevant intestinal protist parasites that affect the human gut, including *Blastocystis* sp., *Cryptosporidium* spp., *Dientamoeba fragilis*, *Entamoeba dispar*, *Entamoeba histolytica*, and *Giardia duodenalis*.

## 2. Materials and Methods

### 2.1. Ethics Statement

The study design and consent procedures of this survey have been approved by the research ethics committee of the Carlos III Instituto de Salud Carlos III (reference number CEI PI17_2017-v3). All human DNA samples were anonymised by using a unique laboratory identifier code, ensuring anonymity and patient confidentiality. This study was conducted in accordance with the principles of the Declaration of Helsinki and those of good clinical practice.

### 2.2. Study Design

This comparative and retrospective observational study is carried out to evaluate the clinical diagnostic performance of five singleplex and one duplex VIASURE Real-Time PCR assays in detecting and differentiating *Blastocystis* sp., *Cryptosporidium* spp., *D. fragilis*, *E. dispar*, *E. histolytica*, and *G. duodenalis* from a panel (*n* = 358) of well-characterized DNA samples.

### 2.3. DNA Reference Panel

A panel of DNA samples that tested positive for *Blastocystis* sp. (*n* = 42), *Cryptosporidium* spp. (*n* = 96), *D. fragilis* (*n* = 37), *E. dispar* (*n* = 11), *E. histolytica* (*n* = 25), *G. duodenalis* (*n* = 115), and other parasitic species of the phyla Apicomplexa (*n* = 14), Euglenozoa (*n* = 8), Microsporidia (*n* = 4), and Nematoda (*n* = 6), were included in the study (Table 1).

Samples of DNA were extracted and purified by applying the QIAamp DNA stool mini kit (QIAGEN, Hilden, Germany) to clinical stool specimens or cultured isolates obtained during routine testing at the Parasitology Reference and Research Laboratory (PRRL) of the Spanish National Centre for Microbiology (SNCM) in Majadahonda in the period 2014–2019. The human samples were taken from patients of all age groups (median age, 10.5 years; standard deviation, 14.9 years; range, 1 to 75 years), although a number of samples were of animal origin, particularly those belonging to species/genotypes adapted to animals or that rarely circulate in humans. All DNA samples were molecularly confirmed by singleplex PCR at the point of initial diagnosis. The singleplex PCR protocols used for the primary detection and differentiation of *Blastocystis* sp., *Cryptosporidium* spp., *D. fragilis*, *E. dispar*, *E. histolytica*, and *G. duodenalis* are fully described in Annexe I of the Appendix A. Where possible, Sanger sequencing was carried out to identify species and genotypes, and all DNA samples were stored at −20 °C until analysis. The complete dataset, including all information about the DNA samples used and the detailed diagnostic results obtained, can be found in Appendix A. The very same DNA reference panel has been previously used to evaluate the diagnostic performance of the CerTest VIASURE Real-Time PCR detection kit in identifying *Cryptosporidium*, *Giardia* and *E. histolytica* [23].

### 2.4. Assays

Five singleplex CerTest VIASURE Real-Time PCR kits used to detect *Cryptosporidium* spp., *D. fragilis*, *E. dispar*, *E. histolytica*, and *G. duodenalis* were evaluated, and a duplex CerTest VIASURE Real-Time PCR kit used to simultaneously detect *Blastocystis* sp. and *D. fragilis* was also included in the study. The main features of the evaluated kits are summarized in Table 2. Each VIASURE Real-Time PCR detection kit includes all the necessary components for the real-time PCR assay (specific primers/probes, dNTPS, buffer and polymerase) in a stabilised format, and an exogenous internal control (EIC) to rule out inhibition of polymerase activity in each well. Assays were performed in strict accordance with the manufacturer’s instructions by using the DT Prime real-time PCR system (DNA Technologies, Moscow, Russia). The used thermal profile included 1 cycle at 95 °C for 2 min, for polymerase activation; followed by 45 cycles at 95 °C for 10 s and 60 °C for 50 s, for denaturation and annealing-extension. To avoid bias, all DNA samples were blindly analysed in triplicate. A sample was considered positive if the obtained cycle threshold (C_T_) value was below 40 and the EIC was positive. Samples with C_T_ values above 40 were considered negative, even with a positive EIC result. A positive (non-infectious synthetic DNA in lyophilized format) and a negative (molecular biology grade water) control (provided in the kit) were used in each run.

### 2.5. Analyses

Cohen’s kappa test was calculated to assess the concordance of diagnostic results obtained with VIASURE Real-Time PCR detection assays and reference singleplex PCR methods used during routine initial diagnostic testing. Cohen’s kappa ranges from zero (no agreement between the two assessors) to one (perfect agreement between the two assessors). A Cohen’s kappa value between 0.81 and 0.99 is considered “near perfect agreement”. Clinical diagnostic sensitivity and specificity, and negative and positive predicted values (with 95% confidence intervals), were calculated by using the free MetaDiSc 1.4 software [24] on the basis of the following formulae:Sensitivity (Se) = [a/(a + c)] × 100
Specificity (Sp) = [d/(b + d)] × 100
Positive predictive value (PPV) = [a/(a + b)] × 100
Negative predictive value (NPV) = [d/(c + d)] × 100
where a = true positive samples, b = false positive samples, c = false negative samples and d = true negative samples. Reference DNA samples that tested positive for *Blastocystis* sp., *Cryptosporidium* spp., *D. fragilis*, *E. dispar*, *E. histolytica*, and *G. duodenalis* that yielded a negative result in the VIASURE assay, were reassessed by using conventional singleplex PCR. DNA samples with a negative result in the VIASURE assay and a positive result in the subsequent confirmatory conventional singleplex PCR were considered to be true false negatives.

To assess the precision of measurements, intra-assay (repeatability) and inter-assay (reproducibility) estimations were carried out with selected stool DNA samples that tested positive for *Blastocystis* sp., *Cryptosporidium* spp., *D. fragilis*, and *E. histolytica* (five each). For intra-assay analyses, five replicates of each stool DNA sample were tested. For practicality purposes, these experiments were conducted by using the multiplex versions of the qPCR assays, including the CerTest VIASURE *Cryptosporidium*, *Giardia*, & *E. histolytica* Real-Time PCR detection kit (Batch: KGE112H-030; expiry date: 1 August 2022) and the CerTest VIASURE *Blastocystis* sp. & *D. fragilis* Real-Time PCR detection kit (Batch: BLD112H-019; Expiry date: 31 March 2025). Validation criteria included the standard deviation (σ) values ≤ 2 and coefficient of variation (CV%) values ≤ 10.3.

## 3. Results

Table 3 shows the agreement of results between the evaluated singleplex/duplex CerTest VIASURE Real-Time PCR detection kits and the reference PCR methods used in the initial diagnosis.

The singleplex VIASURE *Cryptosporidium* spp. assay correctly identified 94.8% (91/96) of the DNA samples that tested positive for this pathogen (92 of human and 4 of wildlife origin, respectively). The assay recognized isolates belonging to six distinct *Cryptosporidium* species, including primarily anthroponotic *C. hominis* (*gp60* subtype families Ia, Ib, and Ie), zoonotic *C. parvum* (*gp60* subtype families IIa, IIc, and IId), canine-adapted *C. canis*, feline-adapted *C. felis*, bovine-adapted *C. ryanae*, and suine-adapted *C. scrofarum* (Appendix A). It is noteworthy that 66.7% (64/96) of samples were monoinfected with *Cryptosporidium* spp., and the remaining were concomitantly infected with *Blastocystis* sp. (19.8%, 19/96), followed by *G. duodenalis* (18.8%, 18/96), *D. fragilis* (8.3%, 8/96) and *E. dispar* (1.0%, 1/96) in nine different combinations (Appendix A).

Regarding *G. duodenalis*, the singleplex VIASURE *Giardia* assay accurately detected 96.5% (111/115) of the DNA samples positive for this protozoon, including zoonotic assemblages A and B, canine-adapted assemblage C, and feline-adapted assemblage F (Appendix A). Overall, 45.2% (52/115) of samples were monoinfected with *G. duodenalis* and the remaining were concomitantly infected by *Blastocystis* sp. (39.1%, 45/115), followed by *D. fragilis* (29.6%, 34/115), *E. dispar* (6.1%, 7/115), and *Cryptosporidium* spp. (4.3%, 5/115), in eight different combinations (Appendix A).

Similarly, the singleplex VIASURE *E. histolytica* assay correctly identified 96.0% (24/25) of the DNA samples positive for this pathogen (Appendix A). Overall, 76.0% (19/25) of samples were monoinfected with *E. histolytica*. All six remaining *Entamoeba*-positive samples were co-infected with *Blastocystis* sp. (24.0%, 6/25) (Appendix A).

The singleplex VIASURE assays for the detection of *E. dispar* and *D. fragilis* correctly identified 100% (11/11 and 37/37, respectively) of the DNA samples positive for these protists (Appendix A). All 11 *E. dispar*-positive samples were co-infected with enteric protist species, including *Blastocystis* sp. (90.9%, 10/11), *D. fragilis* and *G. duodenalis* (36.4%, 4/11 each) and *Cryptosporidium* spp. (9.1%, 1/11), in five different combinations. Out of the 37 *Dientamoeba*-positive samples, 32.4% (12/37) corresponded to monoinfections by protozoan and the remaining were coinfected by *Blastocystis* sp. (59.5%, 22/37), *G. duodenalis* (10.8%, 4/37), and *E. dispar* (8.1%, 3/37), in four different combinations (Appendix A).

With regard to the duplex VIASURE Real-Time PCR for the simultaneous detection of *D. fragilis* and *Blastocystis* sp., the assay correctly identified all (37/37 and 42/42, respectively) DNA samples positive for these protists (Appendix A). This assay performed equally well as the singleplex counterpart in detecting *D. fragilis*. Of the 42 *Blastocystis*-positive samples, 81.0% (34/42) corresponded to monoinfections by this protozoan, and the remainder were coinfected by *G. duodenalis* (16.7%, 7/42), *E. dispar* (4.8%, 2/42), and *D. fragilis* (2.4%, 1/42), in three different combinations (Appendix A).

The singleplex VIASURE assays yielded 10 false-negative results (five *Cryptosporidium* spp., four *G. duodenalis*, and one *E. histolytica*) during testing. Successful EIC amplification of all 10 samples discarded the possibility of PCR inhibition. Reassessment of the five *Cryptosporidium* spp. samples (four *C. hominis*, one *C. parvum*) with the reference PCR method yielded positive results in all five cases, which the VIASURE *Cryptosporidium* assay confirmed to be false-negative results (Appendix A). Reassessments of the four *G. duodenalis* samples and the single *E. histolytica* sample with their corresponding reference PCR methods also yielded positive results (range of C_T_ values: 30.9–41.0) in all cases, and the VIASURE *Giardia* and the VIASURE *E. histolytica* assays confirmed they were false-negative results (Appendix A).

Of the 32 DNA samples used to determine potential cross-reactions, none yielded false-positive results when testing for *Cryptosporidium* spp., *G. duodenalis*, *E. histolytica*, and *E. dispar* (Appendix A). However, four DNA samples cross-reacted with *Blastocystis* sp., including isolates initially positive for *Cystoisospora* sp. (*n* = 1), *Leishmania infantum* (*n* = 1), and *Enterocytozoon bieneusi* (*n* = 2). An additional sample that was initially positive for *E. bieneusi* cross-reacted with *D. fragilis* (Appendix A). In addition, one human and one non-human primate sample harboured co-infections of *G. duodenalis* and *Blastocystis*, respectively, after both were previously detected by initial diagnosis (Appendix A).

Overall, very good agreement (Kappa test values ≥ 0.96) was observed between the results obtained by all VIASURE singleplex/duplex assays and those previously obtained by the reference PCR methods in the initial diagnosis (Table 3).

Taking PCR results obtained during routine initial diagnosis as our reference, we summarize the diagnostic performance of the singleplex/duplex VIASURE assays evaluated here in Table 4. In brief, sensitivity values for the five enteroparasites ranged from 0.94–1.00, specificity values from 0.97–1.00, positive predictive values from 0.89–1.00, and negative predictive values from 0.98–1.00.

Measures of diagnostic precision of repeatability (Table 5) and reproducibility (Table 6) were under acceptable ranges, with the exception of a positive *E. histolytica* sample that failed to be amplified by the CerTest VIASURE *Cryptosporidium*, *Giardia*, & *E. histolytica* Real-Time PCR detection kit.

## 4. Discussion

We carried out a comprehensive evaluation of the diagnostic performance of five singleplex (*Cryptosporidium* spp., *D. fragilis*, *E. dispar*, *E. histolytica*, and *G. duodenalis*) and a duplex (*Blastocystis* sp. and *D. fragilis*) CerTest VIASURE Real-Time PCR assays. *Cryptosporidium* spp., *E. histolytica*, and *G. duodenalis* are the three most clinically relevant intestinal protozoan parasites, and *Blastocystis* sp., *D. fragilis*, and *E. dispar* are common protist species of uncertain pathogenicity.

Most previous similar studies that evaluated commercial qPCR assays used prospectively collected stool samples that were submitted for routine investigation in clinical settings [21,23,25,26,27,28,29,30,31,32,33]. In contrast, this survey purposely used a DNA panel with a wide diversity of parasite species and genotypes, with the aim of providing an evidence-based answer to the question of if the evaluated qPCR assays were suitable for the detection of less common or rare species/genotypes and animal-adapted genetic variants with known zoonotic potential and diverse geographical origins. This is important for intestinal protist species with large genetic diversities, such as *Cryptosporidium* spp., *G. duodenalis*, and *Blastocystis* sp.

Of 50 valid *Cryptosporidium* species [34], 19 have been reported in humans, with *C. hominis* and *C. parvum* accounting for ca. 90% of the human cases of cryptosporidiosis reported globally [35]. The VIASURE *Cryptosporidium* assay was able to detect *C. hominis* (including subtype families Ia, Ib, and Ie), *C. parvum* (including subtype families IIa, IIc, and IId), and animal-adapted *C. canis*, *C. felis*, *C. ryanae*, and *C. scrofarum*. The diagnostic sensitivity and specificity of this assay in the detection of *Cryptosporidium* spp. were 0.94 and 1, respectively. These values were very similar to those obtained with the singleplex and multiplex versions of the assay in previous studies (0.96–1 and 0.99–1, respectively) [23,32]. The sensitivity performance of the VIASURE *Cryptosporidium* assay is also comparable with those (0.96–1) documented for other commercially available qPCR kits, including the EasyScreen Enteric Parasite Detection Kit (Genetic Signatures, Sydney, Australia) and the FilmArray Gastrointestinal Panel (BioFire Diagnostics, Salt Lake City, UT, USA) [21,27]. Lower scores (0.53–0.96) were obtained with the BD MAX Enteric Parasite Panel (Becton, Dickinson and Company, Franklin Lakes, NJ, USA), the FTD Stool Parasites (FAST-Track Diagnostics, Esch-sur-Alzette, Luxembourg), the Gastroenteritis/Parasite Panel I (DIAGENODE, Liège, Belgium), and the RIDA^®^GENE Parasitic Stool Panel II (R-Biopharm AG, Pfungstadt, Germany), assays, respectively [10,25,26,31,32].

*Giardia duodenalis* consists of eight distinct assemblages (A–H) that differ in host specificity, with assemblages A and B being most commonly reported in humans and several other mammal species [36,37]. Sporadic cases of human infection, by canine-adapted assemblages C/D, feline-adapted assemblage F and ungulate-adapted assemblage E, have also been reported, particularly in children and immunocompromised patients [38,39]. The VIASURE *Giardia* assay was able to identify most of the aforementioned *G. duodenalis* assemblages (no D and E isolates were available for testing in this study), demonstrating its usefulness in detecting human and zoonotic genetic variants of the parasite. The diagnostic sensitivity and specificity of The VIASURE *Giardia* assay were 0.94 and 0.99, respectively. More variable results (sensitivity: 0.81–0.97; specificity: 0.94–1) were obtained with the singleplex and multiplex versions of the assay by previous studies [23,32]. It is important to note that other commercially available qPCR assays for the detection of *G. duodenalis* have consistently achieved diagnostic sensitivities over 0.97, including the BD MAX [25,26], the FilmArray [21], and the NanoCHIP Gastrointestinal Panel (Savyon Diagnostics, Ashdod, IL, USA) [29,30] methods. In contrast, the Gastroenteritis/Parasite Panel I has shown poorer performance (0.68–0.76) [31,32].

*Entamoeba histolytica* displays a relatively low level of nucleotide diversity in non-repetitive loci and is not subdivided into intra-species genetic variants for genotyping purposes [40]. This limited genetic variability eases the molecular detection of the parasite. The VIASURE *E. histolytica* assay achieved sensitivity and specificity values of 0.96 and 1, respectively, which is consistent with previous studies, which obtained similar values through the same assay and its multiplex variant (sensitivity: 0.96–1; specificity: 1) [23,32].

Similar, but slightly lower, diagnostic sensitivities (0.92–0.95) have also been reported after the BD MAX [25,26], EasyScreen [21], and Luminex xTAG Gastrointestinal Pathogen Panel (Luminex Molecular Diagnostics, Toronto, ON, Canada) [29] methods were used.

*E. dispar*, which is morphologically identical to *E. histolytica* and closely related to it, is a common finding in human stools that is thought to be avirulent [41], meaning differential diagnosis of *E. histolytica* and *E. dispar* is important in areas where both species occur sympatrically. However, few commercial qPCR-based methods are available for this purpose [32,42], and the VIASURE *E. dispar* assay achieved sensitivity and specificity values of 1, higher than those (sensitivity: 0.95–0.96; specificity: 1) obtained by similar previous studies [32,42].

*Dientamoeba fragilis* isolates also have a very low level of genetic variability, regardless of geographic area of origin or the presence/absence of clinical manifestations. To date, two *D. fragilis* genotypes (1 and 2) have been recognized, with a strong predominance of genotype 1, in humans [43]. The VIASURE *D. fragilis* assay achieved sensitivity and specificity values of 1 and 0.99 (respectively) in both singleplex and duplex versions. This diagnostic performance was in the higher range of what was previously documented in the use of other commercially available qPCR assays (sensitivity: 0.90–0.96; specificity: 1) [32,33].

*Blastocystis* sp. is a frequent protist that colonizes/infects the human gastrointestinal tract. To date, 40 *Blastocystis* subtypes, including ST1-ST17, ST21, and ST23-ST44, are considered taxonomically valid [44,45,46,47,48]. The VIASURE *Blastocystis* + *D. fragilis* assay was able to identify ST1–ST4 (the most frequent STs circulating in humans), in addition to ST5 and ST8. The diagnostic sensitivity and specificity of The VIASURE *Giardia* assay were 0.94 and 0.99, respectively. Overall, these figures were superior to those obtained with other commercial qPCR assays, including the Allplex Gastrointestinal Panel-Parasite assay (Seegene, Seoul, Korea; sensitivity: 0.84–1; specificity: 0.81–082) [49,50] or the LIGHTMIX Gastro Parasite assay (Tib MolBiol, Berlin, Germany; sensitivity: 0.99; specificity: 0.89) [49]. Here it should be noted that the diagnostic performance of the latter assay did not consider the subtype of the isolates investigated.

Direct comparison of results obtained in different studies that evaluate the diagnostic performance of commercial qPCR assays should consider a number of factors, including panel sample size (low sample numbers are likely to result in inaccurate and inconsistent estimates) and composition (genetic diversity of rare or less frequent species/genotypes might affect amplification success rates), sample (reflecting parasite load and sometimes virulence/pathogenicity), and the diagnostic method used as a gold standard.

Part of this study’s advantage over predecessors lies in its careful selection of a large panel of molecularly confirmed DNA samples (PCR and Sanger sequencing) for analysis. This advantage notwithstanding, we are aware that some relevant pathogenic and commensal protozoan species were missing from our panel, including *C. meleagridis* (the third most common cause of cryptosporidiosis in humans) and also potentially cross-reactive species, including *Cyclospora cayetanensis*, *Entamoeba coli*, *Endolimax nana* and *Encephalitozoon intestinalis*, amongst others. Future studies should address this issue of missing species.

## 5. Conclusions

In conclusion, the singleplex and duplex VIASURE Real-Time PCR assays evaluated in the present study provide suitable choices for the molecular diagnosis of *Blastocystis* sp., *Cryptosporidium* spp., *D. fragilis*, *E. dispar*, *E. histolytica*, and *G. duodenalis* during routine clinical practice. Another advantage of these kits is their ready-to-use, stabilised format, which reduces the number of time-consuming steps in the laboratory and allows storage at room temperature. In highlighting the benefits (increased throughput and diagnostic capacity, and reduced response time and improved laboratory workflow) of singleplex and multiplex real-time PCR assays, our data supports their routine use to detect enteric protozoan parasites in laboratory diagnostics.

## Figures and Tables

**Table 1 diagnostics-14-00319-t001:** Panel of laboratory-confirmed DNA samples used in the diagnostic evaluation of the singleplex and duplex CerTest VIASURE Real-Time PCR detection kits.

Phylum	Genus	Species	No. DNA Isolates
Apicomplexa	*Cryptosporidium*	*C. hominis*	73
		*C. parvum*	17
		*C. canis*	1
		*C. felis*	2
		*C. ryanae*	1
		*C. scrofarum*	2
	*Babesia*	*B. divergens*	1
	*Besnoitia*	*B. besnoiti*	2
	*Cystoisospora*	*C. belli*	1
	*Neospora*	*N. caninum*	1
	*Plasmodium*	*P. falciparum*	1
		*P. malariae*	1
		*P. ovale*	1
		*P. vivax*	1
	*Sarcocystis*	*S. arctica*	1
		*S. cruzi*	1
		*S. gigantea*	1
	*Toxoplasma*	*T. gondii*	2
Amoebozoa	*Entamoeba*	*E. histolytica*	25
		*E. dispar*	11
Euglenozoa	*Leishmania*	*L. aethiopica*	1
		*L. amazonensis*	1
		*L. braziliensis*	1
		*L. donovani*	1
		*L. infantum*	1
		*L. major*	1
		*L. mexicana*	1
		*L. tropica*	1
Heterokonta	*Blastocystis*	*Blastocystis* sp.	42
Metamonada	*Giardia*	*G. duodenalis*	115
	*Dientamoeba*	*D. fragilis*	37
Microsporidia	*Enterocytozoon*	*E. bieneusi*	4
Nematoda	*Anisakis*	*A. simplex*	1
	*Dirofilaria*	*D. repens*	1
	*Loa*	*L. loa*	1
	*Mansonella*	*M. perstans*	1
	*Oncocerca*	*O. volvulus*	1
	*Trichuris*	*T. muris*	1
Total			358

**Table 2 diagnostics-14-00319-t002:** Main features of the CerTest VIASURE Real-Time PCR detection kits.

Format and Assay	Protist Species	Targeted Gene	Fluorophore	Batch
Singleplex qPCR				
VIASURE *Cryptosporidium*	*Cryptosporidium* spp.	*ssu* rRNA	FAM	KRYXH-007
VIASURE *D. fragilis*	*Dientamoeba fragilis*	*ssu* rRNA	FAM	DIEXH-007
VIASURE *E. dispar*	*Entamoeba dispar*	*ssu* rRNA	FAM	ETDXH-009
VIASURE *E. histolytica*	*Entamoeba histolytica*	*ssu* rRNA	FAM	ETHXH-010
VIASURE *Giardia*	*Giardia duodenalis*	*ssu* rRNA	FAM	GIAXH-008
Duplex qPCR				
VIASURE *Blastocystis* and *D. fragilis*	*Blastocystis* sp.	*ssu* rRNA	ROX	BLDXH-010
	*Dientamoeba fragilis*	*ssu* rRNA	FAM	

FAM, 6-carboxyfluorescein; ROX, 6-carboxy-X-rhodamine; *ssu* rRNA, small subunit ribosomal RNA.

**Table 3 diagnostics-14-00319-t003:** Direct comparison of the CerTest VIASURE Real-Time PCR detection kits to reference PCR methods used during routine analyses in the initial diagnosis.

Format and Assay	Protist Species	(+/+)	(+/−)	(−/+)	(−/−)	Kappa Test
Singleplex qPCR						
VIASURE *Cryptosporidium*	*Cryptosporidium* spp.	91	0	5	262	0.964
VIASURE *D. fragilis*	*Dientamoeba fragilis*	37	1	0	320	0.985
VIASURE *E. dispar*	*Entamoeba dispar*	11	0	0	347	1
VIASURE *E. histolytica*	*Entamoeba histolytica*	24	0	1	333	0.978
VIASURE *Giardia*	*Giardia duodenalis*	112	0	4	242	0.974
Duplex qPCR						
VIASURE *Blastocystis* and *D. fragilis*	*Blastocystis* sp.	42	4	0	312	0.948
	*Dientamoeba fragilis*	37	1	0	320	0.985

+, Positive result; −, Negative result.

**Table 4 diagnostics-14-00319-t004:** Diagnostic performance of the VIASURE Real-Time PCR detection kits when applied to PCR-confirmed samples during routine analyses in initial diagnosis.

Format and Assay	Overall Agreement	TP	TN	FP	FN	Sensitivity	Specificity	Positive Predictive Value	Negative Predictive Value
Singleplex qPCR									
*Cryptosporidium*	0.986	91	262	0	5	0.94 (0.88–0.98)	1 (0.98–1)	1 (0.96–1)	0.98 (0.95–0.99)
*D. fragilis*	0.986	112	242	0	4	0.96 (0.91–0.99)	0.99 (0.97–1)	0.99 (0.95–1)	0.98 (0.95–0.99)
*E. dispar*	0.99	24	333	0	1	0.96 (0.79–0.99)	1 (0.98–1)	1 (0.85–1)	0.99 (0.98–1)
*E. histolytica*	1.00	11	347	0	0	1 (0.71–1)	1 (0.98–1)	1 (0.71–1)	1 (0.98–1)
*Giardia*	0.985	37	320	1	0	1 (0.91–1)	0.99 (0.98–1)	0.97 (0.86–0.99)	1 (0.98–1)
Duplex qPCR									
*Blastocystis* and *D. fragilis*	0.935	42	312	4	0	1 (0.91–1)	0.98 (0.96–0.99)	0.89 (0.76–0.96)	1 (0.98–1)
	0.985	37	320	1	0	1 (0.9–1)	0.99 (0.98–1)	0.97 (0.86–0.99)	1 (0.98–1)

TP, true positive; TN, true negative; FP, false positive; FN: false negative.

**Table 5 diagnostics-14-00319-t005:** Intra-assay repeatability of the multiplex CerTest VIASURE Real-Time PCR detection kits.

Format and Pathogen	Sample	Mean C_T_	Standard Deviation (σ)	Coefficient of Variation (CV%)
*Cryptosporidium* spp. ^1^	1	31.88	0.31	0.98
	2	31.98	0.52	1.63
	3	31.32	0.43	1.36
	4	31.86	0.58	1.81
	5	31.34	0.39	1.25
*Entamoeba histolytica* ^1^	1	29.10	0.10	0.34
	2	28.78	0.26	0.90
	3	Negative	N/A	N/A
	4	31.36	0.21	0.66
	5	28.22	0.19	0.68
*Giardia duodenalis* ^1^	1	33.66	0.32	0.95
	2	37.20	1.27	3.42
	3	34.18	0.65	1.91
	4	37.80	1.23	3.27
	5	36.20	N/A	N/A
*Blastocystis* sp. ^2^	1	27.76	1.00	3.61
	2	30.04	0.71	2.36
	3	31.04	0.98	3.16
	4	33.56	0.26	0.78
	5	30.98	0.78	2.50
*Dientamoeba fragilis* ^2^	1	22.42	0.59	2.63
	2	21.76	0.19	0.90
	3	20.42	0.48	2.36
	4	23.58	0.36	1.51
	5	20.74	0.78	3.78

^1^ CerTest VIASURE *Cryptosporidium*, *Giardia*, & *E. histolytica* Real-Time PCR detection kit. ^2^ CerTest VIASURE *Blastocystis* sp. & *D. fragilis* Real-Time PCR detection kit. N/A, Not applicable because of insufficient or negative replicate values.

**Table 6 diagnostics-14-00319-t006:** Inter-assay reproducibility of the multiplex CerTest VIASURE Real-Time PCR detection kits.

Format and Pathogen	Sample	Mean C_T_	Standard Deviation (σ)	Coefficient of Variation (CV%)
*Cryptosporidium* spp. ^1^	1	31.48	0.52	1.66
	2	31.66	0.51	1.62
	3	31.12	0.33	1.08
	4	31.32	0.86	2.75
	5	31.30	0.60	1.90
*Entamoeba histolytica* ^1^	1	28.90	0.24	0.85
	2	28.66	0.59	2.04
	3	Negative	N/A	N/A
	4	30.98	0.29	0.95
	5	27.74	0.30	1.07
*Giardia duodenalis* ^1^	1	33.58	0.63	1.86
	2	37.00	0.62	1.67
	3	34.28	1.31	3.83
	4	35.90	0.40	1.11
	5	36.50	0.42	1.16
*Blastocystis* sp. ^2^	1	26.64	0.18	0.68
	2	28.68	0.46	1.61
	3	30.20	1.23	4.07
	4	32.62	1.36	4.18
	5	30.08	1.41	4.70
*Dientamoeba fragilis* ^2^	1	21.44	0.33	1.53
	2	20.88	0.56	2.70
	3	19.28	0.36	1.85
	4	23.28	0.38	1.62
	5	19.98	1.16	5.79

^1^ CerTest VIASURE *Cryptosporidium*, *Giardia*, & *E. histolytica* Real-Time PCR Detection Kit. ^2^ CerTest VIASURE *Blastocystis* sp. & *D. fragilis* Real-Time PCR Detection Kit. N/A, Not applicable because insufficient or negative replicate values.

## Data Availability

The full dataset, including all information about the DNA samples used and the detailed diagnostic results obtained, is in Appendix A.

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
