# Peer review of "Evaluation of the Use of Singleplex and Duplex CerTest VIASURE Real-Time PCR Assays to Detect Common Intestinal Protist Parasites"

_diagnostics, 2024, doi:10.3390/diagnostics14030319_

Round 1

Reviewer 1 Report

Comments and Suggestions for Authors

I think results of comparison between parasite diagnostics techniques is valid, the manuscript can be written putting the more emphasis in the major objective of the work, that is the demonstration of the usefulness of the parasite detection techniques not a advertising.

Author Response

Reviewer #1

I think results of comparison between parasite diagnostics techniques is valid, the manuscript can be written putting the more emphasis in the major objective of the work, that is the demonstration of the usefulness of the parasite detection techniques not a advertising.

Reply: We are unsure what Reviewer #1 means in this comment. In our opinion, this study precisely fulfils the requirement of demonstrating the practical usefulness of the qPCR assays evaluated. To do so, the first step is conducting a thorough evaluation of the diagnostic performance of the methods under investigation against a large and well-characterised panel of DNA samples, allowing the determination of basic parameters such as sensitivity, specificity, positive predictive value, negative predictive value, intra-assay (repeatability) estimation and inter-assay (reproducibility) estimation. This is exactly the task conducted in this study, which concludes that the evaluated methods are well suited for the identification of the pathogens of interest. Another different issue is the direct comparison of the diagnostic performance of these methods against other commercially available assays, or their use in routine clinical practise or in field epidemiological surveys. These tasks should be conducted in independent studies and are out of the scope of the present survey.

Reviewer 2 Report

Comments and Suggestions for Authors

Major Comments

The findings reported in this manuscript exhibit limited novelty. The utility of qPCR for diagnosing common intestinal protozoan infections has been previously demonstrated. Therefore, the novelty of the present work is confined to the evaluation of qPCR test kits with known samples. While this aspect is of interest to those intending to use the kit, I believe that the manuscript requires major revision before it is suitable for publication.

The abstract is inadequate and lacks discussion and a summary. It needs to be revised. Additionally, there is no mention of the advantages of monoplex or duplex VIASURE qPCR when compared to other developed qPCRs for each organism published previously. It would be preferable for the authors to compare the VIASURE qPCR with other widely used, in-house, or conventional techniques in terms of sensitivity, specificity, cost-effectiveness, etc., in the discussion before drawing conclusions, as presented in the manuscript.

Below, we detail some specific issues:

  • Lines 62-65: The benefits of qPCR in i) and ii) are the same as conventional PCR.
  • In Table 1, it is "Cystoisospora," but in line 221, it is "Isospora sp." Please verify and ensure consistency.
  • Italicize the genus and species names, e.g., line 48, 52, 243, 244. Please check and apply throughout the manuscript.
  • In Table 3, please add a footnote to describe the meaning of +/+, +/-, etc.
  • In line 195, change "Entamoeba-positive samples" to "E. dispar-positive samples."
  • Check for typos in line 265.
Comments on the Quality of English Language

English is okay for me. Only some typos need to be fixed.

Author Response

Reviewer #2

The findings reported in this manuscript exhibit limited novelty. The utility of qPCR for diagnosing common intestinal protozoan infections has been previously demonstrated. Therefore, the novelty of the present work is confined to the evaluation of qPCR test kits with known samples. While this aspect is of interest to those intending to use the kit, I believe that the manuscript requires major revision before it is suitable for publication.

Reply: We thank Reviewer #2 for his/her preliminary positive appraisal and for the valuable comments and suggestions. Unfortunately, we disagree with the statement that our study lacks novelty. Please note that this is the first study that formally evaluates the diagnostic performance of five singleplex (for the detection of Cryptosporidium spp., D. fragilis, E. dispar, E. histolytica, and G. duodenalis) and one duplex (for the simultaneous detection of Blastocystis sp. and D. fragilis) CerTest VIASURE Real-Time PCR Kits. In our opinion, assessment of newly commercialized diagnostic kits is an important task that should be always conducted to guarantee that these methods meet the criteria and standards to be used in human diagnostics. Ideally, this task should be conducted by independent institutions such as reference centres and research institutions of recognized prestige. This is precisely the case of the Spanish National Centre for Microbiology, which is the reference centre for infectious agents of viral, bacterial, parasitic, and fungal nature in Spain. Evaluating newly marketed diagnostic kits is one of the multiple activities carried out in our institution taking advantage of the access to a wide range of samples and reference material. The importance of this duty must not be underestimated.

The abstract is inadequate and lacks discussion and a summary. It needs to be revised. Additionally, there is no mention of the advantages of monoplex or duplex VIASURE qPCR when compared to other developed qPCRs for each organism published previously. It would be preferable for the authors to compare the VIASURE qPCR with other widely used, in-house, or conventional techniques in terms of sensitivity, specificity, cost-effectiveness, etc., in the discussion before drawing conclusions, as presented in the manuscript.

Reply: Regarding the abstract, it follows the recommendations of the journal and covers key aspects of the study including design, methods used, main results and conclusions. Please also note that there is a limit of 200 words for the length of the abstract, making little room for improvement. Please note that additional information on the contribution of this manuscript can be found in the conclusion section in current lines 347-355.

Regarding your suggestion of comparing the diagnostic performance of the evaluated methods with other commercially available assays, we agree that this is an important task, but is completely out of the scope of the present study. On this matter, please see our thorough answer to Reviewer #1.

  1. Lines 62-65: The benefits of qPCR in i) and ii) are the same as conventional PCR.

Reply: please note that we are not comparing the advantages of qPCR over conventional PCR, we are just mentioning their main features. It is obvious that some of these advantages are shared between qPCR and PCR methods.

  1. In Table 1, it is "Cystoisospora," but in line 221, it is "Isospora" Please verify and ensure consistency.

Reply: we thank Reviewer#1 for spotting this mistake. The former species “Isospora” has been now replaced by the currently accepted name “Cystoisospora” in current line 219.

  1. Italicize the genus and species names, e.g., line 48, 52, 243, 244. Please check and apply throughout the manuscript.

Reply: Corrected as per requested.

  1. In Table 3, please add a footnote to describe the meaning of +/+, +/-, etc.

Reply: The meaning of the symbols “+” and “–“ are now indicated in the footnote of Table 3 in current line 170.

  1. In line 195, change "Entamoeba-positive samples" to " dispar-positive samples."

Reply: Corrected in current line 194.

  1. Check for typos in line 265.

Reply: We are unable to identify typos in previous line 265.

Round 2

Reviewer 2 Report

Comments and Suggestions for Authors

The authors could adjust the abstract by rephrasing some parts of the results and adding the conclusion.

Author Response

Reviewer #2

The authors could adjust the abstract by rephrasing some parts of the results and adding the conclusion.

Reply: In an attempt to satisfy Reviewer #2, we have modified the abstract adding the requested information in current lines 31-37. Consequently, its length has changed from 216 words in the previous version to 259 words in the current version, exceeding in 59 words the recommended length stated in the Instructions for Authors Guideline.